# Modelling local and general quantum mechanical properties with attention-based pooling

David Buterez [1✉], Jon Paul Janet [2], Steven J. Kiddle[3], Dino Oglic[4] & Pietro Liò[1]

Atom-centred neural networks represent the state-of-the-art for approximating the quantum chemical properties of molecules, such as internal energies. While the design of machine learning architectures that respect chemical principles has continued to advance, the final atom pooling operation that is necessary to convert from atomic to molecular representations in most models remains relatively undeveloped. The most common choices, sum and average pooling, compute molecular representations that are naturally a good fit for many physical properties, while satisfying properties such as permutation invariance which are desirable from a geometric deep learning perspective. However, there are growing concerns that such simplistic functions might have limited representational power, while also being suboptimal for physical properties that are highly localised or intensive. Based on recent advances in graph representation learning, we investigate the use of a learnable pooling function that leverages an attention mechanism to model interactions between atom representations. The proposed pooling operation is a drop-in replacement requiring no changes to any of the other architectural components. Using SchNet and DimeNet++ as starting models, we demonstrate consistent uplifts in performance compared to sum and mean pooling and a recent physics-aware pooling operation designed specifically for orbital energies, on several datasets, properties, and levels of theory, with up to 85% improvements depending on the specific task.

[1] Department of Computer Science and Technology, University of Cambridge, Cambridge CB3 0FD, UK. [2] Molecular AI, Discovery Sciences, R&D, AstraZeneca, Gothenburg 431 50, Sweden. [3] Data Science & Advanced Analytics, Data Science & AI, R&D, AstraZeneca, Cambridge CB2 8PA, UK. [4] Center for AI, Data Science & AI, R&D, AstraZeneca, Cambridge CB2 8PA, UK. ✉email: db804@cam.ac.uk

Geometric deep-learning (GDL) approaches are increasingly used across the life sciences, with remarkable potential and achievements in computational biology (analysing single-cell sequencing data[1,2]), structural biology (prediction of protein structures[3] and protein sequence design[4]), drug discovery[5] and simulating rigid and fluid dynamics[6] being only a few examples. The simple but powerful formulation of GDL methods such as graph neural networks (GNN) motivated the investigation of long-standing problems from a new perspective, particularly in fields such as computational chemistry where the GDL abstractions can be naturally applied to objects like atoms and molecules (nodes and graphs), as well as their interactions (edges).

Approximating quantum mechanical properties using machine learning (ML) is of significant interest for applications in catalysis, material and drug design[7,8]. However, traditional physics-based methods are severely limited by computational requirements that scale poorly with system size[9]. In the pursuit of accurate, scalable, and generalisable ML models, several different strategies have been proposed. So-called semiempirical methods[10,11], such as the Neglect of Diatomic Orbital Overlap (NDDO[12]) and modern related methods[13], tight-binding DFT (xTB[14]), or 'quantum force fields'[15], can greatly improve the scaling of traditional quantum chemistry methods through simplification or approximation of the underlying physics, without restrictions on the domain of applicability imposed by purely data-driven methods. Nonetheless, there is great interest in supporting this vision by taking a purely data-driven approach and developing quantum machine learning (QML) models based on accurate physical methods such as density functional theory (DFT) combined with large and diverse collections of data (e.g., QM9[16], QMugs[17], nablaDFT[18] and QM7-X[19]). Another approach is to devise transfer learning datasets and algorithms that can extract useful patterns from less accurate, but cheaper and more scalable simulations that ultimately benefit predictions at a higher fidelity level[8,17,20,21]. At the same time, advances in GNN architectures and the ability to exploit specific features of quantum data such as atom positions and directional information such as bond angles and rotations are active areas of research that have produced state-of-the-art models[22–25].

Most of the GNN advances have focused on more expressive ways of defining atom (node) representations and local interactions in increasingly large neighbourhoods centred on each of the nodes ('k-hops'). For example, SchNet[22] starts with atom embeddings based on the atom type. These initial representations are then processed by atom-wise layers which are implemented as linear transformations. The atom-wise layers are also combined with convolutional layers that satisfy rotational invariance to form interaction blocks. The interaction blocks are modular elements and can be stacked for a more expressive architecture. DimeNet[23] formulates the task as a message-passing exercise, while also introducing directionality by considering the angles between atoms. Instead of atom embeddings, DimeNet computes directional embeddings between pairs of atoms $j, i$ that incorporate atomic distances and angles by aggregating other embeddings directed towards the source atom $j$. The construction also guarantees invariance to rotations. Furthermore, instead of using raw angles, DimeNet represents distances and angles through a spherical 2D Fourier-Bessel basis, a physics-inspired decision that was also empirically found to be preferable. The original DimeNet architecture was subsequently updated to a faster and more accurate model denoted DimeNet++ by replacing costly operations with fast and expressive alternatives[24]. Recently, GDL architectures that are invariant to translations, rotations, and reflections such as E(n) GNNs have proven competitive in the prediction of quantum mechanical properties[25].

Even with the accelerated development of QML methods and the heterogeneity of recent approaches, a common element for most QML models is that they naturally operate at the level of atom representations, for example through message-passing steps. However, many prediction targets of interest are formulated at the molecular level, e.g., total energy, dipole moment, highest occupied molecular orbital (HOMO) energy, lowest unoccupied molecular orbital (LUMO) energy, etc. Thus, an aggregation scheme must be used to combine the atom representations into a single molecule-level representation. This task is typically handled with simple fixed pooling functions like sum, average, or maximum. Despite their appealing simplicity, there are growing concerns regarding the representational power of this class of functions[26,27]. In the following section, we also discuss the concurrently developed orbital-weighted average (OWA), a physics-based method designed specifically for orbital properties and which also seeks to improve upon the standard pooling operators by exploiting the local and intensive character of the target property[27]. Buterez et al. also highlighted the lacklustre performance of standard pooling functions in a variety of settings, particularly on challenging molecular properties[28]. As an alternative to standard pooling, the authors proposed replacing the fixed functions with learnable functions implemented as neural networks. When applied to conventional message passing architectures (GCN[29], GAT[30] and GATv2[31], GIN[32], PNA[33]) that operate on the molecular graph with node features extracted from the SMILES[34] representation, neural pooling functions provided significant uplifts in performance and faster convergence times.

Apart from expressive power, the standard pooling functions are also widely used thanks to being permutation invariant with respect to the order of the atom representations that are being aggregated. Furthermore, these simple operations are also usually aligned with fundamental physical principles. For example, the total energy, a molecular property, can be obtained as the sum of the atom energies. In general, molecular properties that scale linearly with the number of atoms can be well approximated by fixed functions such as sum or average. However, it is not uncommon for the target property to behave non-linearly or be localised around a small subset of atoms which determines its value. Bioaffinity (the achieved level of inhibition or activation of a drug-like molecule against a protein target) is a property where we can reasonably expect that most of the effect comes from an active group of atoms[28]. In QML, a canonical example of a property that can be delocalised as well as localised is the HOMO energy, since it corresponds to a specific molecular orbital that may involve contributions from multiple atomic orbitals.

The attention mechanism that is now ubiquitous in deep learning thanks to its success in a wide range of fields, including the life sciences[35–38], can be considered a natural step forward in the search for expressive aggregation functions. In this work, we investigate the use of an attention-based pooling function on atomistic systems for the prediction of general and localised quantum properties with state-of-the-art 3D-coordinate aware models (the high-level workflow is illustrated in Fig. 1). The chosen design satisfies a collection of desirable features, some of which were previously mentioned, namely (i) permutation invariance with respect to node (atom) order, (ii) increased representational power compared to standard pooling operators thanks to the underlying neural networks, (iii) the ability to model arbitrary, potentially long-range or localised relationships due to the attention mechanism, and (iv) generality and simplicity; the proposed method is applicable to any molecular property (including quantum properties), and can be used as a drop-in replacement on any architecture that uses traditional pooling methods without any modifications to the model itself. Attention-

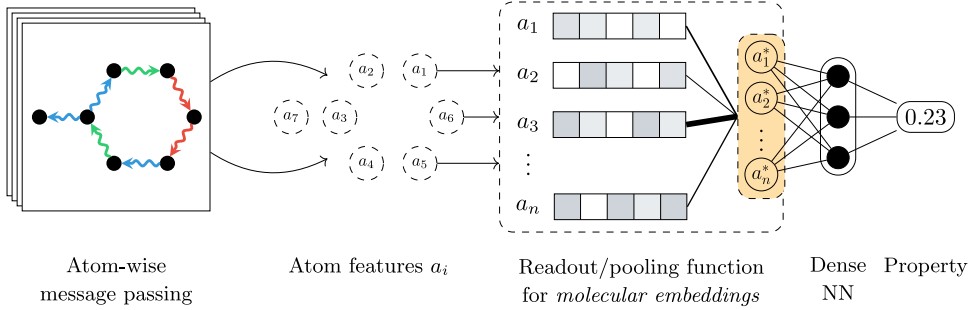

**Fig. 1 A high-level overview of attention-based pooling for atom-centred networks.** A common step in most atom-centred neural networks is the aggregation or pooling of learnt atom features into a molecule-level representation through a dedicated function (highlighted with a dashed box). Traditionally, simple functions that satisfy permutation invariance such as sum, mean, or maximum are used for this step. Alternatively, a more expressive molecular representation can be computed by neural networks, for example using attention to discover the most relevant atomic features.

based pooling is also applicable to equivariant message-passing architectures (e.g., refs. [25,39]).

The current work represents an extension to previous work which considered only 2D molecular graphs but demonstrated the potential of attention-based pooling for predicting properties such as the HOMO energy[28]. Previous work considered attention-based pooling for GNNs in a general setting, including molecular and non-molecular data. For molecular data, atom representations were generated from the SMILES representation using RDKit[40] and covered properties such as the atomic number, formal charge, the hybridisation type, number of bonded hydrogens, and other similar properties. We did not use edge (bond) features. Learning occurs by message passing according to the graph connectivity and the specific operator (e.g., GCN, GIN, GAT, PNA, etc.). Readout functions that leverage attention showed considerable uplifts, particularly for molecular tasks such as bioaffinity and a wide range of quantum properties. However, state-of-the-art architectures for modelling quantum properties leverage, at the very least, the 3D atomic coordinates available for each molecule, with more advanced models also considering directional information such as the angles between bonds. Hence, the potential benefits of attention-based pooling remain unexplored for this class of widely used neural networks.

Here, we demonstrate consistent uplifts in performance (as measured by the mean absolute error—MAE) compared to sum and mean pooling, chosen as representatives of the established methods, on a selection of standard datasets of different sizes and simulated at different levels of theory, including QM7b (7211 molecules)[41–43], QM8 (21,786 molecules)[44,45], QM9 (130,831 molecules)[16,44], QMugs (665K molecules)[17], and the CCSD and CCSD(T) datasets from MD17 (1500 molecules for Aspirin, Benzene, Malonaldehyde, Toluene, and 2000 for Ethanol)[46].

The chosen target properties ensure a diverse selection of tasks, also enabling comparisons with previous work. QM7b covers a range of molecular properties, from extensive atomisation energy to intensive excitation energies, frontier orbital properties and polarizability. These provide a comprehensive set of properties corresponding to the stability and reactivity of molecules. Frontier orbital properties (LUMO, HOMO) are examples of localised properties, i.e., specific atoms are involved in the definition of these orbitals - even if their orbitals can be spread over large portions of the molecule in some cases, in contrast to constitutional energies, which are inherently functions of all atoms. QM8 provides ground state and excitation energies at different levels of theory. For QM9 and QMugs, being larger datasets, we selected frontier orbital properties (HOMO and LUMO) as representative properties to investigate, and also for comparison with the OWA method which is derived specially for orbital properties. For QMugs, we additionally include the global extensive total energy.

While the frontier orbital localisation can vary quite a bit and in some extreme cases be a large part of the structure, it still has a defined spatial extent and that corresponds to a subgraph of the original nodes. In fact, OE62 contains molecules where the spatial extent of the HOMO has been studied[27]. This is contrastable with total energy, which involves every atom in all cases.

Sum pooling is the default choice in many implementations (e.g., SchNet, DimeNet), usually outperforming mean and maximum pooling[26] or matching them[28]. We also evaluate the proposed methods against OWA on the OE62[47] dataset and conclude that attention-based pooling can match and outperform OWA depending on the configuration (e.g., number of attention heads). While our method introduces a large number of learnable parameters, this is normal for a standard attention implementation and does not significantly affect training times or introduce overfitting, as discussed later in *Results*.

Overall, we conclude that attention-based pooling (ABP) is an excellent drop-in replacement for standard readout functions in atom-centred neural networks. ABP is general and applicable to any quantum property, in most cases outperforming existing methods, and is particularly suitable for localised properties. ABP incurs a small computational resource penalty that can be further reduced on modern hardware.

## Methodology

**Pooling functions.** We start by assuming an atomistic model that operates on positional inputs (distances, angles, etc.) and computes individual representations that require aggregation into a single molecule-level embedding. The specifics of the architecture or the implementation do not matter as long as the assumptions hold. For example, many message-passing neural networks can be summarised into the following generic formulation that computes node-level features $\mathbf{h}_a$[48]:

$$\mathbf{h}_a = \phi\left(\mathbf{x}_a, \underset{v \in \mathcal{N}_a}{\oplus} \psi(\mathbf{x}_a, \mathbf{x}_v)\right) \qquad (1)$$

where $a, v$ are nodes (atoms), $\mathbf{x}_i$ denotes atom representations, $\mathcal{N}_i$ is the 1-hop neighbourhood of atom $i$, $\oplus$ is a node-level aggregation function such as sum or average, and $\phi, \psi$ are learnable functions such as multi-layer perceptrons (MLPs). It should be noted that there are many variations and extensions of Eq. (1), and this is only an example of a possible architecture. Importantly, once the message passing steps or equivalent updates are done, the atom-level representations are aggregated into a molecule-level representation $\mathbf{h}_m = \oplus_{i \in \mathcal{V}}(\mathbf{h}_i)$, where $\mathcal{V}$ is the collection of atoms in the molecule (note that this $\oplus$ can be different from the one in Eq. (1)). This operation is often called an *aggregation*, *pooling*, or *readout* function.

**Attention-based pooling**. To design an expressive pooling function that considers the entire context of the molecule (i.e., all computed atom representations) and is self-contained (does not require any additional inputs), we leverage the existing Set Transformer framework introduced by Lee et al. for set modelling[49], and proposed by Buterez et al. for use in node aggregation[28]. In other words, the pooling operation is reframed as a set summarisation task, with an output that corresponds to the desired molecular embedding. This is achieved by assembling building blocks defined using a standard multihead attention mechanism:

$$\text{Attention}(Q, K, V) = \omega(QK^{\mathsf{T}})V \tag{2}$$

$$\text{MultiHeadAttention}(Q, K, V) = \text{Concatenate}(H_1, ..., H_m)W^O \tag{3}$$

$$\text{where } H_i = \text{Attention}(QW_i^Q, KW_i^K, VW_i^V) \tag{4}$$

The standalone attention module Attention$(\cdot, \cdot, \cdot)$ receives input *query*, *key*, and *value* vectors, of dimension $d_k$, $d_k$, and $d_v$ respectively (queries and keys have the same dimension) and gathered in matrices $Q, K, V$, respectively. With $\omega(\cdot) = \text{softmax}(\cdot/\sqrt{d_k})$, the attention operation computes a weighted sum of the values where a large query-key dot product assigns a larger weight to the corresponding value. In multihead attention, $Q, K, V$ are projected to new dimensions by learnt projections $W_i^Q, W_i^K, W_i^V$, respectively, for a total of $m$ independent times. The results are processed by an attention module, with the concatenated output attention *heads* being projected with $W^O$.

Following the original Set Transformer implementation, we use the lower-level multihead and self-attention blocks (MABs, respectively SABs) and pooling by multihead attention (PMA) to define an encoder-decoder architecture that embeds input atom vectors into a chosen dimension $d$, then learns to aggregate or compress the encoded representations into a single vector, the *molecule representation*. The encoder is defined as

$$\text{MAB}(X, Y) = H + \text{Linear}_\phi(H) \tag{5}$$

$$\text{where } H = X + \text{MultiHeadAttention}(X, Y, Y) \tag{6}$$

$$\text{SAB}(X) = \text{MAB}(X, X) \tag{7}$$

$$\text{Encoder}(X) = \text{SAB}^n(X) \tag{8}$$

with a decoder:

$$\text{PMA}_k(Z) = \text{MAB}\left(S_k, \text{Linear}_\phi(Z)\right) \tag{9}$$

$$\text{Decoder}(Z) = \text{Linear}_\phi\left(\text{SAB}^n\left(\text{PMA}_k(Z)\right)\right) \tag{10}$$

Here, Linear$_\phi$ denotes a linear layer followed by an activation function $\phi$, SAB$^n(\cdot)$ represents $n$ consecutive applications of SABs, and $S_k$ is a collection of learnable $k$ seed vectors that are randomly initialised (PMA$_k$ outputs $k$ vectors). The resulting Set Transformer module can be used as a pooling function by following two sequential steps. Firstly, by processing all the atomic representations into features $Z$ with the encoder. The decoder is then tasked with transforming the features into a single-vector representation of the set. Here, we refer to this pooling function as attention-based pooling (ABP):

$$\text{ABP}(X) = \text{Decoder}(\text{Encoder}(X)) \tag{11}$$

**Orbital-weighted average pooling**. Chen et al.[27] have concurrently observed that the standard pooling functions (sum, average, maximum) might not accurately describe physical properties that are highly localised and intensive, such as orbital properties, and in particular the HOMO energy. Instead, they discuss the importance of pooling functions that can attribute different weights or 'importance' for a subset of atomic representations. For example, the softmax function softmax$(\epsilon_1, ..., \epsilon_n) = \sum_{i=1}^{n} \frac{\exp(\epsilon_i)}{\sum_{j=1}^{n} \exp(\epsilon_j)}$, where $\epsilon_i$ are atomic representations of an $n$-atom system that in this case are assumed to be scalars. The general form is given by weighted average (WA) pooling:

$$f_{\text{WA}} = \sum_{i=1}^{n} w_i \epsilon_i \tag{12}$$

where additionally we assume that the learnable weights $w_i$ are normalised by softmax to sum to 1.

From a physical perspective, the weights that the neural network will learn for HOMO energy prediction should tend towards the orbital coefficients $l_i$ that describe the fraction of the orbital that is localised on a given atom $i$. To incorporate this idea into the pooling function, Chen et al.[27] propose the following strategy:

1. Pre-compute the orbital coefficients for the dataset (offline)
2. Use a separate atomistic model to learn the weights for $f_{\text{WA}}$, which are forced to be close to the pre-computed coefficients by an updated loss function:

$$\mathcal{L}_{\text{OWA}} = \frac{1}{n_{\text{train}}} \left[ \alpha \sum_{A=1}^{n_{\text{train}}} \left( E_{\text{HOMO}}^A - \sum_{i=1}^{n_A} w_{(A,i)} \epsilon_{(A,i)} \right)^2 + \beta \sum_{A=1}^{n_{\text{train}}} \sum_{i=1}^{n_A} \left( l_{(A,i)} - w_{(A,i)} \right)^2 \right] \tag{13}$$

where $n_{\text{train}}$ denotes the number of training systems $A$ in the dataset, $n_A$ is the number of atoms in a system $A$, $E_{\text{HOMO}}^A$ is the target HOMO energy for a system $A$, and $\alpha$ and $\beta$ are global parameters that indicate the relative contributions of orbital energies and localisations to the loss. The resulting pooling function with the learnt weights is denoted by $f_{\text{OWA}}$ (orbital-weighted average).

### Design and implementation

As stated in *Methodology*, the proposed attention-based pooling function can be applied to a variety of atomistic modelling algorithms. Here, we chose to evaluate our methods using two architectures: SchNet and DimeNet++, which were briefly discussed in the introduction. Both models are widely known and used, making them easily accessible in general-purpose deep-learning libraries such as PyTorch Geometric (used here)[50,51]. Furthermore, DimeNet++ is a particularly competitive model which outperforms both contemporary and newer models (e.g., E(n) GNNs[25]).

For our evaluation, we chose sum pooling as a representative of the standard pooling methods. It is the default choice for SchNet and DimeNet++, and in our previous extensive evaluation of graph pooling functions, we did not observe significantly better performance for any of the three functions on bioaffinity tasks[28]. Furthermore, from a physical perspective sum pooling can be considered a natural choice for approximating certain quantum properties. Nonetheless, we also report results for mean pooling in Supplementary Table 3.

Here, we have used the PyTorch Geometric implementations of SchNet and DimeNet++, modified to support attention-based pooling. As of PyTorch Geometric version 2.3.0 (available at the

time of writing), the proposed pooling function is natively available as `SetTransformerAggregation` (based on our implementation). Unless otherwise noted, we use relatively deep models to ensure that the atom-level representations learnt before pooling are expressive enough. In particular, we use SchNet models with 256 filters, 256 hidden channels (hidden embedding size), and 8 interaction blocks, and otherwise default parameters (total parameter count before pooling: 2.3 million), and DimeNet++ models with 256 hidden channels (hidden embedding size), 6 interaction blocks, an embedding size of 64 in the interaction blocks, a basis embedding size of 8 in the interaction blocks, an embedding size of 256 for the output blocks, and otherwise default parameters (total parameter count before pooling: 5.1 million). The models chosen here are larger than the defaults in PyTorch Geometric and the original DimeNet study[23]. In addition, while it is common to output scalar representations for the atoms, we keep the same dimension for the atom representations as used inside the models (before output), i.e., 256. This ensures that the attention-based pooling can benefit from the full representation, although in principle it is possible to apply it to scalars. We follow the output of attention-based pooling with a small MLP, obtaining a scalar prediction.

We evaluate our methods on all the properties of the QM7b and QM8 datasets, on HOMO and LUMO energy prediction for QM9 and QMugs, as well as on energy prediction tasks from MD17, which provide a challenging setting due to the limited amount of data. We have also considered total energy prediction for QMugs as an example of a non-local property on the largest and most diverse of the available datasets. Results are provided for both SchNet and DimeNet++, with sum and attention-based pooling, i.e., 4 results per (property, dataset) pair. A batch size of 128 is used for all models. To ensure an accurate and self-contained comparison, we randomly generate 5 different train, validation, and test splits for each dataset using a ratio of 80%/10%/10%, and report the average MAE ± standard deviation. The MAE is used as it is widely used in the literature to evaluate atomistic models on quantum property prediction. Since ABP introduces several hyperparameters compared to standard functions, we evaluate a small set of common hyperparameter choices and select the best configurations according to the validation set.

We also compare attention-based pooling with OWA on the OE62 dataset used by Chen et al.[27] on both HOMO and LUMO energy. For this comparison, we use the provided OWA source code as a starting point, and use the same underlying SchNet implementation provided by the `schnetpack` 1.0.1 library[52], including the same SchNet hyperparameters (embedding size of 128, 128 filters, 64 Gaussian functions, 6 interaction blocks, a cutoff radius of 5, followed by 4 atom-wise layers with an input of 128 features). A batch size of 40 was used, as larger models would run out of memory when using hardware equipped with 32GB of video memory. For attention-based pooling, we modified the atom-wise component to output representations of the same dimensionality as the inputs, as described above. We also generated 5 random splits using the same ratio as Chen et al.[27] (32,000 molecules for train, 19,480 for validation, and 10,000 for test), and report both the MAE and the RMSE (root mean squared error).

## Results

Our results indicate that attention-based pooling outperforms sum pooling on the majority of datasets and quantum properties, including properties computed at different levels of theory (Fig. 2 and Table 1). Training and validation metrics are reported in Supplementary Tables 1 and 2. On QM7b, using ABP on top of SchNet results in an average decrease (across all tasks) in MAE of 50.5%, with the highest decrease on the 'Polarizability (self-consistent screening)' task (85.13%). The smallest decrease in MAE is observed for 'Atomisation energy (DFT/PBE0)' (23.98%). When using DimeNet++, there is a more modest average decrease in MAE of 14.64%, with the most improved task being 'Atomisation energy (DFT/PBE0)' (58.41%). The only QM7b task where ABP does not improve performance is 'Maximal absorption intensity (ZINDO)' (−0.37%) when using DimeNet++. The ABP-based DimeNet++ models generally match the ABP-based SchNet models, suggesting that we are reaching the performance ceiling for these model configurations and tasks.

On QM8, there is an average decrease in MAE of 19.23% across all tasks for SchNet, and all tasks are improved when using ABP. The most improved task is 'E2-PBE0/def2SVP' (25.07%), and the least is 'f2-CC2' (13.15%). When using DimeNet++, the average decrease in MAE due to ABP is of 2.31%, with the most improved task being 'E1-CAM' (7.69%). We observed slightly worse performance when using ABP for only two tasks: 'f2-CAM' (−1.11%) and 'f2-PBE0/def2TZVP' (−0.48%). In general, for both SchNet and DimeNet++, the least improved tasks when using ABP involve the oscillator strength $f_2$.

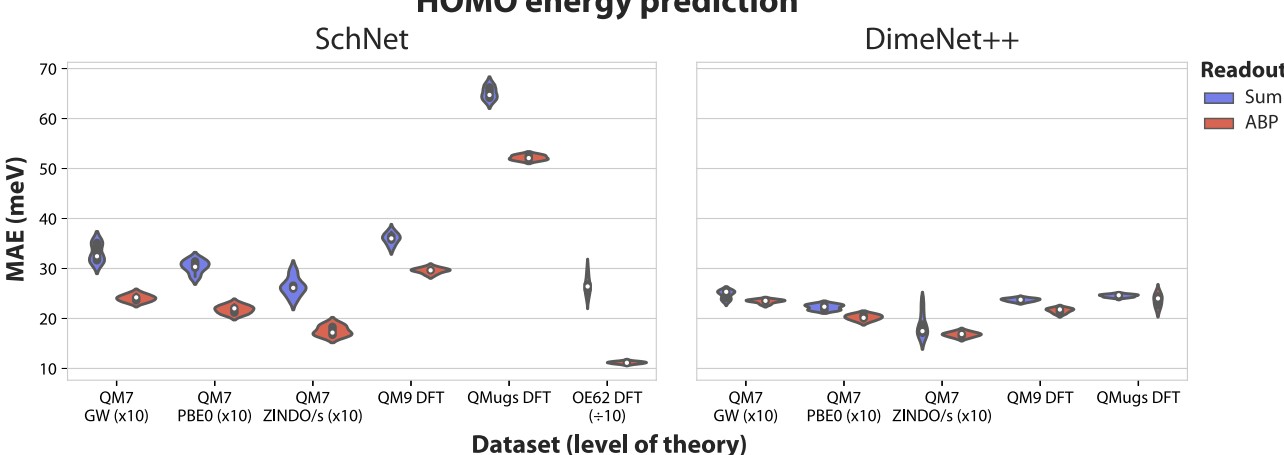

**Fig. 2 Performance of attention-based pooling versus sum pooling on HOMO energy prediction tasks.** SchNet and DimeNet++ models evaluated on HOMO energy prediction on different datasets (QM7b with different levels of theory, QM9, and QMugs), with the mean absolute error reported on test sets corresponding to five different random splits for each dataset. The metrics of some datasets are scaled by 10 to ensure a similar scale for all datasets (indicated by `(x10)' or `(÷10)'). The exact metrics are reported in Tables 1 and 2.

**Table 1 Test set results for pooling methods on a wide range of datasets and tasks.**

| QM7b | SchNet | | DimeNet++ | |
|---|---|---|---|---|
| **Task – level of theory (unit)** | **Sum** | **ABP** | **Sum** | **ABP** |
| Atom. energy – ZINDO/s (meV) | 3.8594 ± 1.5259 | **3.1129 ± 0.8027** | 4.2682 ± 0.8056 | **2.6945 ± 0.4007** |
| Electron affinity – ZINDO (meV) | 2.0445 ± 0.2396 | **1.2602 ± 0.1764** | 1.3579 ± 0.0739 | **1.2362 ± 0.0568** |
| Exc. energy at MA – ZINDO (meV) | 40.1961 ± 2.0175 | **29.9163 ± 1.1184** | 32.3594 ± 0.6809 | **29.6964 ± 0.6585** |
| First exc. energy – ZINDO (meV) | 2.4291 ± 0.1244 | **1.4933 ± 0.0635** | 1.5712 ± 0.1302 | **1.4225 ± 0.0492** |
| HOMO – GW (meV) | 3.3114 ± 0.1898 | **2.405 ± 0.0653** | 2.4745 ± 0.0851 | **2.3425 ± 0.0368** |
| HOMO – PBE0 (meV) | 3.0388 ± 0.1313 | **2.1782 ± 0.078** | 2.2241 ± 0.0555 | **2.022 ± 0.0546** |
| HOMO – ZINDO/s (meV) | 2.6389 ± 0.1907 | **1.7517 ± 0.0985** | 1.8492 ± 0.2358 | **1.6779 ± 0.0462** |
| Ion. potential – ZINDO/s (meV) | 4.1453 ± 0.3259 | **2.6968 ± 0.1829** | 3.1309 ± 0.2063 | **2.6465 ± 0.1521** |
| LUMO – GW (meV) | 3.1983 ± 0.3127 | **2.201 ± 0.1799** | 2.2671 ± 0.1794 | **2.1901 ± 0.1435** |
| LUMO – PBE0 (meV) | 2.0418 ± 0.1531 | **1.4993 ± 0.052** | 1.6098 ± 0.1364 | **1.4843 ± 0.0918** |
| LUMO – ZINDO/s (meV) | 1.7833 ± 0.1637 | **1.0079 ± 0.0914** | 1.022 ± 0.0834 | **0.9466 ± 0.0611** |
| MA intensity – ZINDO (arbitrary u.) | 0.0621 ± 0.0043 | **0.0499 ± 0.0036** | **0.0504 ± 0.0027** | 0.0506 ± 0.0025 |
| Polarizability – DFT/PBE0 ($Å^3$) | 0.0545 ± 0.0131 | **0.0313 ± 0.0023** | 0.0414 ± 0.0072 | **0.0339 ± 0.0058** |
| Polarizability – SCS ($Å^3$) | 0.0402 ± 0.0102 | **0.0217 ± 0.0017** | 0.0376 ± 0.0067 | **0.0286 ± 0.0027** |
| **QM8** | | | | |
| E1-CAM (meV) | 2.5281 ± 0.1053 | **2.0662 ± 0.0543** | 2.0174 ± 0.1371 | **1.8733 ± 0.0418** |
| E1-CC2 (meV) | 2.9611 ± 0.0892 | **2.4858 ± 0.0577** | 2.356 ± 0.0874 | **2.2682 ± 0.0453** |
| E1-PBE0/def2SVP (meV) | 2.7438 ± 0.0984 | **2.2815 ± 0.0326** | 2.2133 ± 0.0924 | **2.1114 ± 0.0681** |
| E1-PBE0/def2TZVP (meV) | 2.811 ± 0.1223 | **2.3284 ± 0.0476** | 2.198 ± 0.0604 | **2.121 ± 0.0411** |
| E2-CAM (meV) | 3.6754 ± 0.0665 | **2.9817 ± 0.0661** | 2.9757 ± 0.1292 | **2.9061 ± 0.0635** |
| E2-CC2 (meV) | 4.7751 ± 0.3393 | **3.9243 ± 0.0775** | 3.8655 ± 0.1533 | **3.8595 ± 0.1065** |
| E2-PBE0/def2SVP (meV) | 3.9925 ± 0.1757 | **3.1921 ± 0.038** | 3.2751 ± 0.1098 | **3.1367 ± 0.0773** |
| E2-PBE0/def2TZVP (meV) | 3.8727 ± 0.1213 | **3.1896 ± 0.0757** | 3.1725 ± 0.0666 | **3.1336 ± 0.0631** |
| f1-CAM (meV) | 8.8874 ± 0.7468 | **7.5481 ± 0.4352** | 6.7815 ± 0.3782 | **6.7211 ± 0.4862** |
| f1-CC2 (meV) | 10.1161 ± 0.2368 | **8.5145 ± 0.612** | 7.7868 ± 0.2583 | **7.7728 ± 0.3966** |
| f1-PBE0/def2SVP (meV) | 8.5 ± 0.5847 | **7.2271 ± 0.3896** | 6.6124 ± 0.56 | **6.4726 ± 0.4533** |
| f1-PBE0/def2TZVP (meV) | 8.3752 ± 0.5421 | **7.309 ± 0.5944** | 6.9194 ± 0.2837 | **6.5475 ± 0.4608** |
| f2-CAM (meV) | 21.171 ± 0.8133 | **17.7844 ± 0.6103** | **16.1078 ± 0.7989** | 16.2888 ± 0.5456 |
| f2-CC2 (meV) | 25.0288 ± 0.8254 | **22.1203 ± 0.6011** | 20.6984 ± 1.0388 | **20.4243 ± 1.1357** |
| f2-PBE0/def2SVP (meV) | 19.1633 ± 1.1406 | **16.329 ± 1.1011** | 14.9302 ± 0.7617 | **14.9229 ± 0.693** |
| f2-PBE0/def2TZVP (meV) | 19.6788 ± 0.4903 | **17.0492 ± 1.0804** | **15.4548 ± 0.7086** | 15.5292 ± 0.7807 |
| **QM9** | | | | |
| HOMO – DFT (meV) | 35.9852 ± 1.0706 | **29.5771 ± 0.5172** | 23.7112 ± 0.3027 | **21.5772 ± 0.4591** |
| LUMO – DFT (meV) | 33.5047 ± 0.8853 | **26.96 ± 0.8733** | 20.8319 ± 0.6665 | **20.4879 ± 0.5223** |
| **QMugs** | | | | |
| HOMO – DFT (meV) | 65.0941 ± 1.2428 | **52.1722 ± 0.475** | 24.536 ± 0.26 | **23.6096 ± 1.2122** |
| LUMO – DFT (meV) | 62.0224 ± 0.83 | **47.3707 ± 0.9157** | 21.4916 ± 0.5581 | **20.9528 ± 0.3756** |
| Total Energy – DFT ($E_h$) | 9.0511 ± 2.8475 | **3.7143 ± 1.3582** | 3.4022 ± 1.8096 | **2.7445 ± 1.6937** |
| **MD17 (energies)** | | | | |
| Aspirin – CCSD (kcal/mol) | 3.9347 ± 0.1281 | **3.414 ± 0.0582** | 2.5373 ± 0.0609 | **2.327 ± 0.0738** |
| Benzene – CCSD(T) (kcal/mol) | 0.4863 ± 0.2946 | **0.297 ± 0.029** | 0.2899 ± 0.0339 | **0.2665 ± 0.0552** |
| Ethanol – CCSD(T) (kcal/mol) | **0.6761 ± 0.0349** | 0.6848 ± 0.0272 | 0.5757 ± 0.026 | **0.5667 ± 0.009** |
| MDA – CCSD(T) (kcal/mol) | 0.9369 ± 0.0822 | **0.897 ± 0.0279** | 1.0579 ± 0.0262 | **1.0484 ± 0.0356** |
| Toluene – CCSD(T) (kcal/mol) | 1.5291 ± 0.4943 | **0.95 ± 0.0483** | 1.1317 ± 0.0461 | **1.1035 ± 0.0541** |

Test MAE (mean ± standard deviation from 5 data random splits) for QM7b, QM8, QM9, QMugs, and MD17 (best MAEs and table headers in bold).
*MA* maximal absorption, *SCS* self-consistent screening, *atom.* atomisation, *exc.* excitation, *ion.* ionisation, *u.* units, *MDA* malondialdehyde.

For the larger quantum datasets (QM9 and QMugs), we train and evaluate models for the HOMO and LUMO energy prediction tasks. For HOMO, on QM9 we notice decreases in MAE of 21.67% and 9.89% for SchNet and DimeNet++, respectively. On QMugs, the decreases are of 24.77% and 3.92% for SchNet and DimeNet++, respectively. Similar uplifts are observed for LUMO. Interestingly, total energy prediction on QMugs is improved by a large amount (58.97%) on SchNet, and by a moderate amount on DimeNet++ (19.34%), despite not being a local property like the HOMO or LUMO energies.

For the small MD17 datasets, we observe a decrease in MAE for Aspirin of 15.46% and 9.04% for SchNet and DimeNet++, respectively, for Benzene of 63.71% and 8.79%, for Ethanol of −1.26% and 1.57%, for Malonaldehyde of 4.44% and 0.91%, and

for Toluene of 60.95% and 2.56%. For these smaller datasets, we used SchNet with 128 filters, 128 hidden channels, and 4 interaction blocks, and DimeNet++ models with 128 hidden channels, 4 interaction blocks, and an embedding size of 128 for the output blocks. For the cases where using ABP did not improve the MAE (e.g., SchNet on Ethanol), we noticed that different underlying architectures can help improve upon the sum pooling result (usually more complex models with more interaction blocks).

To further validate the performance of ABP compared to sum pooling for each algorithm (i.e., SchNet and DimeNet++ in Table 1), we performed Wilcoxon signed-rank tests as the data is not normally distributed according to scipy's normaltest function : $p = 9.65 \times 10^{-16}$ (SchNet sum), $p = 2.12 \times 10^{-16}$

**Table 2 Test set results for pooling methods and HOMO energy prediction on the OE62 dataset (SchNet).**

| Readout | MAE | RMSE | # ABP parameters |
|---|---|---|---|
| Sum | 0.2656 ± 0.0177 | 0.4032 ± 0.0168 | N/A |
| Average | 0.1437 ± 0.0016 | 0.2043 ± 0.0009 | N/A |
| OWA | 0.1135 ± 0.0019 | 0.1670 ± 0.0021 | N/A |
| ABP(64, 4, 2) | 0.1158 ± 0.0020 | 0.1697 ± 0.0021 | 995,456 |
| ABP(64, 8, 2) | 0.1130 ± 0.0019 | 0.1660 ± 0.0028 | 3,694,720 |
| ABP(64, 16, 2) | **0.1119 ± 0.0022** | 0.1655 ± 0.0045 | 14,205,056 |
| ABP(64, 16, 3) | 0.1124 ± 0.0006 | **0.1648 ± 0.0015** | 18,403,456 |

Test MAE and RMSE (mean ± standard deviation from 5 data random splits) for SchNet-based HOMO energy prediction on the OE62 dataset, including the number of learnable parameters for the attention-based pooling (ABP). The ABP configuration is reported as 'ABP(*embedding size, number of attention heads, number of SABs*)'. The number of learnable parameters for the underlying SchNet model (not including the readout) is 480,002. The smallest MAE/RMSE values and table headers are highlighted in bold. The unit used for energy is eV, as used by Chen et al.[27].

**Table 3 Test set results for pooling methods and LUMO energy prediction on the OE62 dataset (SchNet).**

| Readout | MAE | RMSE |
|---|---|---|
| Sum | 0.1654 ± 0.0083 | 0.2374 ± 0.0097 |
| Average | 0.1393 ± 0.0059 | 0.2037 ± 0.0097 |
| OWA | 0.1281 ± 0.0030 | 0.1858 ± 0.0056 |
| ABP(64, 8, 2) | 0.1050 ± 0.0024 | 0.1630 ± 0.0047 |
| ABP(64, 16, 2) | **0.1010 ± 0.0011** | **0.1580 ± 0.0033** |

Test MAE and RMSE (mean ± standard deviation from 5 data random splits) for SchNet-based LUMO energy prediction on the OE62 dataset. The smallest MAE/RMSE values and table headers are highlighted in bold. The unit used for energy is eV, as used by Chen et al.[27]. The naming conventions and numbers of parameters are reported in Table 2.

(SchNet ABP), $p = 8.19 \times 10^{-14}$ (DimeNet++ sum), $p = 2.73 \times 10^{-13}$ (DimeNet++ ABP). The Wilcoxon tests indicated statistical significance for SchNet ($p = 1.14 \times 10^{-6}$) and DimeNet++ ($p = 6.69 \times 10^{-6}$).

In contrast to sum pooling, mean pooling performed similarly but generally slightly worse for the majority of properties (Supplementary Table 3) for both SchNet and DimeNet++. HOMO and LUMO energy prediction is generally improved when using mean pooling, especially for larger datasets (QM9, QMugs). For example, when using SchNet on QM7b for HOMO energy prediction, mean pooling leads to a decrease in MAE, on average, of 15.44% (GW), 13.85% (PBE0), and 8.55% (ZINDO/s), while for DimeNet++ we noticed an *increase* in MAE by 5.1% (GW), 0.75% (PBE0), and a decrease of 0.44% for ZINDO/s. Mean pooling decreased the MAE for the QM9 HOMO task by 9.75% for SchNet and 2.26% for DimeNet++, and by 16.57% for SchNet and 0.83% for DimeNet++ on the LUMO energy prediction task. However, ABP achieves lower prediction error than both sum and mean pooling.

When compared to OWA pooling for HOMO energy prediction (Table 2), attention-based pooling matches or even slightly outperforms OWA depending on the ABP configuration, despite not leveraging pre-computed orbital coefficients. This can be observed both in terms of RMSE (the metric chosen by Chen et al.) and MAE (used throughout the rest of the paper). Here, we also study LUMO energy prediction which is not considered by Chen et al.[27], but is available in the OE62 dataset. We find that OWA offers a smaller improvement with respect to average pooling on LUMO energy prediction (8.04%) compared to HOMO energy (21.02%), as given by the MAE, with similar trends for the RMSE (Table 3). Furthermore, whereas for HOMO the attention-based pooling offered a small but noticeable improvement compared to OWA, for LUMO we observe a more significant improvement of 21.16% for ABP.

When not using orbital coefficients for a dataset such as QM7b (Table 4) where they are not readily available, we find that weighted average pooling still outperforms sum pooling by a noticeable amount (around 10%); however, ABP improves even further, with decreases in MAE between 41% and 57%.

**Resource utilisation and scaling.** Since attention-based pooling is a relatively new direction for neural networks, it is important to characterise the amount of resources used in addition to the underlying model (here, SchNet and DimeNet++). The main implementation of ABP that is used throughout the paper exhibits quadratic time and memory scaling in the number of atoms of a molecule, as the standard attention mechanism computes pairwise scores between all atom representations outputted by SchNet or DimeNet++. Currently, all molecules have to be padded with zeros to the maximum number of atoms per molecule in the used dataset, since otherwise the tensor dimensions would not match across different molecules. Although recent versions of PyTorch (2.0+) support ragged tensors where the dimensions might not match, the implementation is still an early prototype at the time of writing and not all tensor operations required by ABP are available. It is also worth highlighting that inside ABP, the SABs can be trivially replaced with induced self-attention blocks (ISABs), an alternative that computes attention between the original set of size $n$ and a small set of learnable parameters of size $m$ (typically $m \leq 16$), scaling as $n \cdot m$ in terms of memory instead of quadratically in $n$. We have not investigated the potential impact of this approximation on the predictive performance.

We have recorded the time spent per epoch (using PyTorch Lightning's `Timer` callback function) and the maximum amount of allocated memory using PyTorch's `torch.cuda.max_memory_allocated` function, for SchNet and DimeNet++ models with sum and ABP readouts in Fig. 3 (QM9, QMugs) and Supplementary Fig. 1 (QM7b, QM8). On QM9, the model with the largest ABP readout (16 attention heads and 64 embedding dimensions per head) adds, on average, 27.57 seconds of training per epoch for SchNet (79.54% relative to sum pooling) and 14.05 (9.77% relative) for DimeNet++. On QMugs, ABP added 11 minutes and 37 seconds for SchNet (101.75% relative) and 9 minutes and 43 seconds (18.33% relative) for DimeNet++. Despite ABP increasing the training time per epoch by similar amounts (e.g., around 10 min for QMugs), the relative change for SchNet is considerably larger since SchNet is 4 to 5 times faster than DimeNet++ in our experiments.

Unlike the time spent training, the available memory is a fixed quantity and is thus often a more important consideration. We observe that ABP only increases the amount of allocated memory by 0.63 GB for QM9 when using SchNet (37.47% relative to sum pooling) and by 0.43 GB for DimeNet++ (4.60% relative). On QMugs, there is an increase of 2.75 GB for SchNet (28.82% relative) and 2.49 GB for DimeNet++ (9.19% relative). As can also be seen in Fig. 3, the majority of the allocated memory is for

**Table 4 Test set results for pooling methods including weighted average pooling for HOMO energy prediction on QM7b.**

| QM7b | SchNet | | |
|---|---|---|---|
| Task (level of theory) | Sum | WA | ABP |
| HOMO (GW) | 3.311 ± 0.190 | 2.951 ± 0.112 | **2.342 ± 0.037** |
| HOMO (PBE0) | 3.039 ± 0.131 | 2.748 ± 0.174 | **2.022 ± 0.055** |
| HOMO (ZINDO/s) | 2.639 ± 0.191 | 2.353 ± 0.163 | **1.678 ± 0.046** |

Test MAE for the QM7b dataset (HOMO energies) including WA pooling, which does not use pre-computed orbital coefficients (best MAEs and table headers in bold). The energy unit is meV.

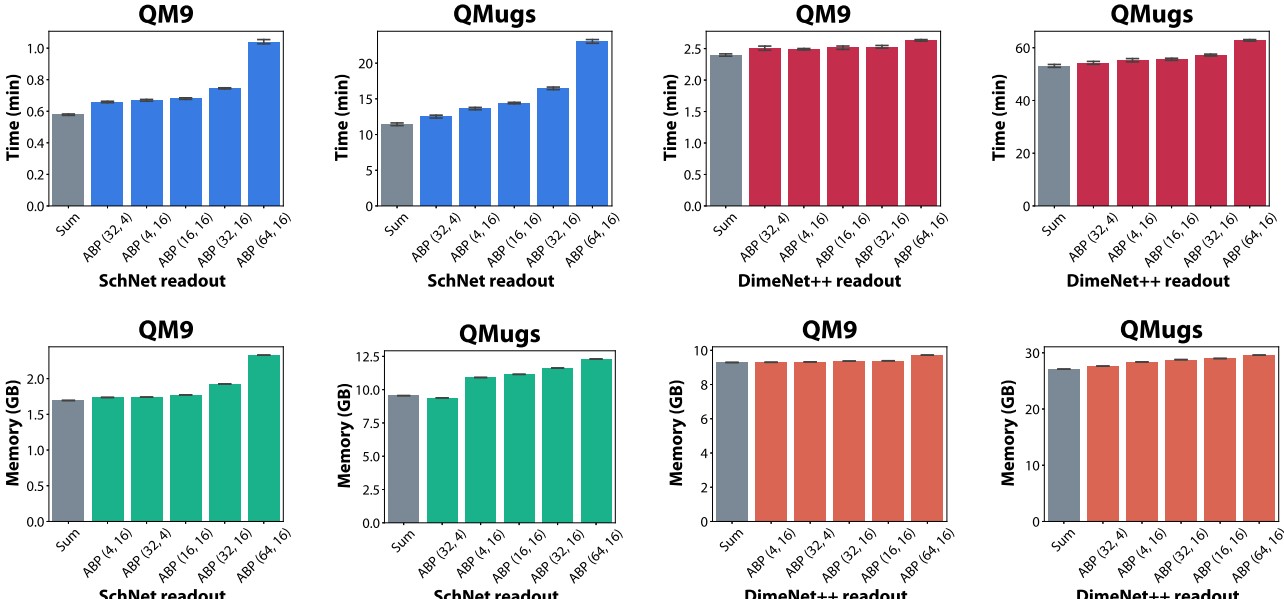

**Fig. 3 Analysis of the time and memory consumption of the ABP and sum readouts on all datasets.** Time spent training per epoch (minutes) and memory consumption (GB) of SchNet and DimeNet++ models with sum and ABP readouts (standard attention) for different datasets. Several configurations of ABP (hidden dimension per attention head, number of attention heads) are included for comparison, and all ABP readouts use 2 self attention blocks (SABs). Results are reported for 10 different runs of the same model configuration on a single NVIDIA V100 GPU with 32GB memory. QMugs models use a batch size of 64, while all other models use a batch size of 128. Error bars represent a 95% confidence interval.

the underlying SchNet or DimeNet++ model, with only a minor penalty associated with ABP.

Despite the quadratic time and memory scaling of ABP, it is important to note that the first limiting factor when training large models is the underlying model. We were unable to run DimeNet++ on QMugs (the largest molecule has 228 atoms) with the same batch size (128) as for the other datasets even when using standard readouts (sum). Even with a batch size of 64, the model used 27.10 GB and almost hit the memory limit of professional-grade NVIDIA V100 graphics processing units with 32GB of memory (Fig. 3). DimeNet exhibits quadratic scaling within some molecular operations[53], which can exacerbate the time and memory utilisation for datasets with large and complex molecules such as QMugs.

Although the standard attention implementation used throughout the paper can be considered memory efficient and even time efficient for models such as DimeNet++, recently many alternatives have been proposed that scale sub-quadratically with the number of atoms and with severalfold reductions in training time, without relying on approximations. Such techniques include FlashAttention[54,55] (linear in sequence length for natural language processing applications—here, in the number of atoms) and memory efficient attention[56] (scales with the square root of the sequence length and it has been tested with sequences of over 1 million tokens). We have used the official FlashAttention implementation (https://github.com/Dao-AILab/

**Table 5 Time and memory utilisation of a sum readout DimeNet++ model compared to an ABP implementation leveraging the recent FlashAttention method ('ABP-Flash') on the QMugs dataset and a single NVIDIA RTX 3090 GPU with 24GB memory.**

| Readout | Time (min) | Memory (GB) |
|---|---|---|
| Sum | 23.0816 ± 0.055 | 15.2246 ± 0 |
| ABP-Flash(64, 16, 2) | 25.3882 ± 0.0732 | 15.4809 ± 0.0011 |
| ABP-Standard(64, 16, 2) | N/A | OOM |

The sum readout DimeNet++ model is also considerably faster than in Fig. 3 thanks to the newer GPU architecture and half-precision training (the wide dynamic range `bfloat16` tensor type was used), which generally halves the training time and memory consumption. Standard attention ('ABP-Standard') is not applicable ('N/A') in this case as a model with the same settings does not fit within the memory of a single RTX 3090 GPU (out-of-memory, 'OOM'). Results are reported for 5 different runs of the same model configuration. Table headers are presented in bold.

flash-attention), which is currently experimental and only optimised for the most recent hardware architectures, to replace the standard attention mechanism used within ABP (Table 5). Differently from the previous measurements, we now use an NVIDIA RTX 3090 graphics card to fully benefit from the latest architectural advancements. At the time of writing, the

FlashAttention implementation is only available for half-precision floating point numbers, which can generally already halve the memory consumption. We further use an 8-bit Adam optimiser as a further modern optimisation to reduce memory usage[57,58]. Overall, the increase in memory cost for the largest ABP readout is just 1.68% compared to sum pooling (Table 5), and the training time increase per epoch is reduced to just 2 minutes and 18 seconds (9.99% relative). Although using half-precision floating point numbers might reduce stability when training, this is generally addressable by improvements in the prototype implementations of attention and techniques such as gradient clipping[59]. This demonstrates that advances in the implementation of attention methods stand to alleviate most of the increased computational complexity associated with ABP.

## Discussion

The presented results suggest that attention-based pooling is preferable to sum-based pooling for intensive, localised properties such as HOMO and LUMO energy prediction. Perhaps less expected is the uplift in performance on other properties that are not as localised as the HOMO and LUMO energies, for example, total energy prediction on QMugs or energy prediction on the small MD17 datasets. In this latter case, it is also remarkable that a data-driven method like ABP is able to often outperform standard pooling when training with around 1,000 data points. Apart from the physical motivation behind improving localised property prediction, it should be noted that the attention mechanism adds an additional layer of expressivity to the network, enabling better approximation of general-purpose properties. Moreover, although we have presented the 'main' network such as SchNet or DimeNet++ and the pooling function as separate components, they work synergistically, especially for highly expressive learnable and differentiable pooling functions. It is not unlikely that the patterns learnt at the pooling level can propagate to the main model, leading to an improved, holistic model behaviour.

We have also observed that there is generally a large discrepancy in performance between SchNet and DimeNet++ when using sum pooling, while the performance difference between the two models when using ABP tends to be lower, particularly for the smaller datasets. One possible explanation for the performance difference between sum pooling models is that DimeNet++ is significantly more expressive (in terms of representational power) compared to SchNet. Similarly, the models used in our previous work only considered information derivable from the SMILES encoding, but not the 3D atomic coordinates. Such models are not competitive with models that leverage this information, like SchNet. DimeNet++ adds a further layer of complexity by considering directional information that is not directly accessible within SchNet (also note that representations based on spherical Bessel functions and spherical harmonics were empirically found to be more helpful than the raw angles in DimeNet++). The fact that ABP SchNet and DimeNet++ models are close in performance, especially for smaller datasets such as QM7b and QM8, suggests that attention can, to a certain extent, extract more information from the input data by working synergistically with the underlying atom-centred model and increasing the overall expressivity. However, for large and complex datasets such as QMugs, the explicit incorporation of directional information appears necessary for high performance.

When compared to OWA on the diverse OE62 dataset for HOMO energy prediction (Table 2), attention-based pooling can match and even outperform it depending on the attention configuration (i.e., number of attention heads and hidden dimensions). Under the same conditions, we observe an over 20% improvement for ABP on LUMO energy prediction (Table 3).

This is an interesting conclusion as OWA requires the pre-computation of orbital coefficients and their explicit incorporation in the loss function. Since this additional information is not required by our method, it suggests that most of the information that is required to reach this level of performance is already available in the network, but it is not fully exploited. Interestingly, Chen et al.[27] noticed that WA can occasionally outperform OWA with the actual orbital coefficients, most likely due to the increased network flexibility. Thus, models that deviate from or even omit physical references can sometimes be preferable.

Although OWA is an innovative and physically based approach, its scalability and applicability might limit its full potential. The method requires the pre-computation of orbital coefficients, which are not generally available for most published datasets. Furthermore, the OWA weights are learnt by a second atomistic model which is inherently not scalable as it imposes a doubling of the model's requirements and an additional model must be added for each new property to be predicted in a multi-task scenario. Perhaps most limiting, the OWA approach is engineered specifically for orbital properties, with no straightforward analogue for properties that do not have a well-defined orbital basis.

We illustrate this last point by considering HOMO energy prediction tasks on the QM7b dataset without pre-computing the orbital coefficients (Table 4). The OWA method thus takes the more general, non-orbital specific WA form of Eq. (12). The results indicate that attention-based pooling outperforms WA by about the same margin as WA pooling outperforms sum pooling. We also take the opportunity to highlight the similarity between the (O)WA methods and the well-known Deep Sets framework that considers sum-decomposable functions, where individual items are processed by simple neural networks such as MLPs before being summed[60]. Although Deep Sets offer a theoretically sound construction, our work has previously suggested and exemplified that Deep Sets-style pooling does not match the performance of attention-based pooling[28].

Overall, the replacement of simple pooling functions with an attention-based pooling function (here, the Set Transformer) has empirically proven to be the optimal choice in the majority of evaluated settings. Attention is particularly well-suited for tasks involving non-linear or localised patterns, although it is often effective for properties of different natures. In theory, an expressive and permutation-invariant module such as the Set Transformer can also learn to represent functions like sum, average, or maximum if necessary, although the amount and quality of data also becomes a consideration in such a scenario. Practically, the proposed pooling function acts as a drop-in replacement for existing pooling operations and does not require any pre-computations or modifications to the underlying network. Although the standard attention mechanism that we used here has quadratic time and memory scaling, we did not observe significantly larger training times or prohibitive increases in consumed memory for any of the shown experiments. We also did not notice overfitting or divergence due to the large number of parameters.

## Data availability

All the datasets used throughout the paper are publicly available through different hosting services, as indicated in the main text. For ease of use, we provide pre-processed versions of certain datasets which are accessible by following the instructions included in the source code. The numerical source data for Figs. 2 and 3 (also Supplementary Information 2) are provided as Supplementary Data 1, respectively Supplementary Data 2.

## Code availability

The source code that enables all experiments to be reproduced is hosted on GitHub: https://github.com/davidbuterez/attention-based-pooling-for-quantum-properties.

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

## Acknowledgements
We are thankful for the Ph.D grant and access to the Scientific Computing Platform within AstraZeneca.

## Author contributions
D.B., J.P.J., S.J.K., D.O., and P.L. were involved in conceptualising and reviewing the manuscript. D.B. designed and carried out experiments and produced the figures and tables. J.P.J., S.J.K., D.O., and P.L. jointly supervised the work.

## Competing interests
D.B. has been supported by a fully funded Ph.D grant from AstraZeneca. J.P.J., S.J.K., and D.O. are employees and shareholders at AstraZeneca. All other authors declare no competing interests.
