## [Peer Review File · Communications Chemistry]

Reviewers' comments:

Reviewer #1 (Remarks to the Author):

The present study investigates the use of a learnable pooling function that leverages an attention mechanism to model interactions between atom representations, and is demonstrated to provide improved performance relative to sum pooling and a recent physics-aware pooling operation designed specifically for orbital energies. Overall, the method appears promising, and represents an important advance in the field with potential for broad application.

There are several aspects of the manuscript which could be made more clear and the scientific content improved before publication can be recommended. These are summarized below:

1. In the introduction, the authors oversimplify the general strategies that have been taken for developing accurate, scalable and generalizable machine learning potentials. The ANI potential (ref. 8) and QM1 (ref. 9) models are mentioned as examples of the two classes that are highlighted. A couple of points that bear mention: fast, approximate QM methods such as NDDO-based semiempirical models or density-functional tight-binding methods can be formulated to have favorable (linear) scaling properties using linear-scaling electronic structure methods, or so-called quantum mechanical force field frameworks, e.g., *J. Phys.: Condens. Matter* (2017) 29, 383002-383016. This enables new QM/ML potentials to be developed that are more robust, do not have to be explicitly trained to avoid highly non-physical regions of configurational space, and that have certain known physics, such as long-ranged electrostatics, built into the models (it should be noted that the ANI potential of reference 8 does not, and breaks down when used for systems with varying charge). A drug discovery example is given in *J. Chem. Phys.* (2023) 158, 124110. While these points do not detract from the advances reported in this work, they do warrant placing into better context in the introduction.

2. The distinction between the current work and that of reference 21 should be better clarified beyond the statement "The current work represents an extension to previous work which considered only 2D molecular graphs, but demonstrated the potential of attention-based pooling for predicting properties such as the HOMO energy [21]."

3. It would be valuable to have tables in the SI that break down MAEs separately for training, validation and testing sets.

4. To my understanding the architecture depicted in Fig. 1, which utilizes all atomic information in a system, is designed mainly for a small molecular systems, but may not be transferable to large systems. It would be valuable for the authors to please clarify this.

5. In Section 5, the author claims that "Although the standard attention mechanism that we used here has quadratic time and memory scaling, we did not observe significantly larger training times or prohibitive increases in consumed memory for any of the shown experiments." Is there any data to support this conclusion? Also, it is helpful to readers to give a detailed increased percentage.

6. In Eq.2, what does \top (T) represent? If \top is transpose, “T” is usually used to represent transpose, not \top .

7. In Eq. 13, the meaning and usage of symbol α and β are not explained.

8. “Chen et al.” appears many times in the main text as well as the table captions, but the citation only appears when it appears the first time, making it hard to find this citation – it would be useful to have it listed at each occurrence.

Reviewer #2 (Remarks to the Author):

The manuscript introduces a learnable pooling function aimed at improving the prediction of localized physical properties of interest. Compared to the recently developed physics-aware model, OWA, carefully designed for predicting these properties, attention-based pooling methods demonstrate comparable or even superior performance. Unlike the OWA model, this method does not require precomputed orbital coefficients resulting in computational less intensity. In addition, including a pooling layer enhances expressivity and synergistic interplay with the main networks, helping the prediction of less localized properties as well. This makes the proposed model more suitable for general usage. This work addresses a crucial aspect of machine learning applications in predicting various physical properties, and the manuscript is well-written. However, the manuscript can be further improved by addressing the following concerns.

1. The authors state that the proposed attention-based method does not significantly impact training times, which may hold true for the datasets considered in this work. However, it is important to note that atomic center-based machine learning potentials or machine learning force fields are generally designed to handle large size-extensive systems, allowing for simulations involving thousands to millions of atoms. Attention-based methods can potentially introduce additional computational costs for such large system, since quadratic scaling with the number of atoms N is problematic. It would be beneficial for the authors to address this aspect in the paper and provide a scaling test in terms of number of atoms present in system at least in inference stage, considering their potential impact on computational efficiency for larger-scale systems.

2. The authors have provided a description of the datasets and level of theory used in this study in the last paragraph of the introduction and results sections. While this information is crucial, it is equally important to discuss the choice of target properties and provide a more detailed elaboration on them. Since the aim of this work is to investigate local and general quantum mechanical properties, this will enhance the clarity and completeness of the manuscript.

3. It would be valuable to further discuss the discrepancy improvement on DimeNet++ and ScheNet achieved by using Attention-Based Pooling (ABP) compared to sum pooling (in Figure 2). It is worth noting that in Chen et al. (DOI: 10.1039/d3sc00841j), a comparable improvement on GAP and ScheNet was shown in Figure 3 by using Ordered Weighted Averaging (OWA) pooling methods against sum

pooling. Discussing and contrasting these findings can provide a comprehensive understanding of the interplay between proposed ABP and different machine learning models.

4. The prediction of tensorial properties, e.g., dipole moment and molecular spectra, is another class of “general” properties in physics (arXiv:2102.03150, arXiv:2202.01449). Previous studies relied on equivariant representations and gated equivariant blocks to predict these properties. I suggest the authors should discuss the use of attention-based pooling in this direction and how it can be incorporated into equivariant neural networks as well.

5. It would be informative to describe the technical differences and challenges between the current work and previous studies focusing on 2D molecular graphs.

(<https://openreview.net/forum?id=yts7fLpWY9G>.)

Reviewer #3 (Remarks to the Author):

This manuscript presents a learnable pooling function for atomistic neural network to predict quantum mechanical properties. The attention-based pooling methods have been developed in recent years for graph neural networks based on 2D molecular graphs. This work employs attention-based pooling to 3D coordinate-based approaches. This method is compared to sum and other aggregation functions in many benchmark datasets, and it is found to give better performance. Overall, this work contributes to the recent interest for atom-to-molecule aggregation in atomistic and graph neural networks and it deals with a timely topic. I would recommend it for publication, however I have few suggestions to improve it.

1) The title named ‘local and general quantum mechanical properties’ is unclear. In principle, some local properties still belong to general properties. Furthermore, this paper defined HOMO energy as a representative of localized properties. This is only half true. HOMO orbitals could be localized but also delocalized in these QM9, QM7 test datasets.

2) The authors explained the reason why they only focused on sum pooling. Nevertheless, I would highly suggest the authors to test the average pooling function as well based on extensive/intensive properties. Sum pooling is physically understandable for extensive properties in SchNet, which is based on local atomic interactions. However, the average pooling is commonly used for intensive properties even in the SchNet paper (J. Chem. Phys. 2018,148, 241722), where the HOMO energies in QM9 were calculated by using average pooling instead of sum pooling. Sum pooling usually leads to unphysical results for intensive properties. Although the authors claim that, based on their early work, the sum and average pooling have no big difference for quantum mechanic properties prediction, these tested properties are however almost extensive properties (such as U0,U298,CV,ZPVE,R2 refer to Comput. Chem. Eng. 2023,172.108202), which could make no big difference for sum/average pooling. This paper tested some intensive properties, the average pooling is therefore of high interest to compare for intensive properties, especially those in Figure 2 and Tables 1 and 4.

3) Some long sentences are difficult to understand. For example, ‘SchNet starts with atom embeddings based on the atom type and atom-wise layers implemented as linear transformations that are combined with convolutional layers satisfying rotational invariance’. This sentence means that the SchNet starts

with atom embeddings based on atom type and atom-wise layers. However, the atom embedding is followed by the atom-wise layer instead of simultaneously processing. The other example sentence from the paper is 'The resulting Set Transformer module can be used as a pooling function by encoding ...'.

4) For the Methodology part, please carefully check the equations. The parameter 'hu' seems to be 'ha' since u has never been introduced in this part. 'of dimension dk, dk, and dv' should be 'dq, dk, and dv'.

5) The authors compared the ABP performance with OWA pooling. The RMSE of SchNet average pooling and OWA pooling in OE62 dataset are slightly higher than the results of Chen et al. (Chem. Sci., 2023, 14, 4913). One possible reason is perhaps that 1500 epochs used in this work overfitted the models. Please check the validation loss as the function of epoch to avoid overfitting.

6) The OWA method incorporates the physical information in the model to make the model physically understandable. It can predict orbital energies as well as orbital locations simultaneously, which is useful for excitation and charge transfer studies of organic semiconductors. Other properties can also be predicted based on different physical references. Different properties should use different physical references for the atomic weights. Such as for LUMO energy prediction, LUMO orbital coefficients should be used for the model training. Using the HOMO orbital information model for the prediction of LUMO energies is doable but lacks physical sense.

7) This learnable pooling function has a huge number of parameters, which could result in a demand for big data. It is interesting to investigate the learning curve of the APB performance with the increase of the training data compared with the sum or average pooling functions or the application for small dataset.

8) The authors advertise their pooling method and mentioned that this pooling can deal with long-range interaction. Long-range interactions are challenging for current the state-of-the-art ML models. However, this should be tested such as in molecular dynamic simulations for some weak interaction systems like proteins. I would assume that the authors over advertise their method a bit as there is no test in the paper and it is not clear how this attention-based pooling could solve such problem.

9) The reference should be updated. There are some chemrxiv papers already published in journals.

10) This attention pooling method depends on the attention configuration (i.e. number of attention heads and hidden dimensions). This raises a question, how efficiently gets the optimal parameters and how expensive it is compared to the non-learnable pooling methods such as sum/average or other pooling functions.

We would like to thank the editor and the reviewers for considering our manuscript and for the insightful suggestions. These contributions have enabled us to deliver a more comprehensive analysis and to take steps towards a highly efficient implementation which should facilitate the adoption of the proposed techniques by a wider audience.

We identified a number of common observations within the reviews, mostly regarding the scalability and resource consumption of the proposed attention-based pooling (ABP), but also the relationship with previous work and considering other standard pooling functions such as the mean.

We have revised the manuscript and addressed all of the reviewers' suggestions, with the largest additions being a '*Resource utilization and scaling*' subsection within *Results*, including a new figure (Figure 3), and additional metrics and results within the new Supplementary Information. We also provide a point-by-point response below, starting with a common section dedicated to the resource utilization suggestions that were common in the reviews.

Resource utilization and scaling

Since ABP is a relatively new approach, we agree that it is important to characterize its training time, memory consumption, and scalability to larger and more complex data. We have added an extensive discussion in the revised manuscript, and here we reproduce the main results and conclusions:

1. Asymptotic complexity, training time, and memory consumption

The standard attention mechanism discussed so far scales quadratically in terms of time and memory with respect to the number of atoms (graph nodes) in a molecule. However, we have observed that this complexity leads to relatively minor increases in consumed memory and non-disruptive increases in training times (measured per epoch). This is likely due to the nature of the data; in our evaluation, the largest molecules are encountered in the QMugs dataset: 228 atoms (note that all inputs have to be padded to the maximum number of atoms). In Natural Language Processing, standard attention has been successfully applied to sequence lengths of up to 512 tokens (the sequence length is the equivalent of the number of atoms in a molecule for our tasks) [1].

For the most resource intensive model (DimeNet++) and the largest dataset, both in terms of number of molecules and number of atoms per molecule (QMugs), ABP adds 2.49 GB of consumed memory, on top of the 27.10 GB consumed by DimeNet++ itself (an increase of 9.19%). In terms of training time, ABP added 9 minutes and 43 seconds to the 53 minutes and 4 seconds taken by DimeNet++ (an increase of 18.33%).

Generally, the training times and memory utilization of ABP for SchNet increase by the same absolute amount (e.g. around 10 minutes on QMugs). However, since SchNet is 4 to 5 times faster on the same hardware, the relative increase in training time compared to standard pooling is more substantial (close to double).

Overall, we note that we were able to train ABP models on the same hardware and similar timescales as models with standard readout functions.

2. More efficient attention

Motivated by recent advances in efficient attention implementations, we also investigated a recent technique termed FlashAttention which has linear scaling in terms of memory and severalfold improvements in training (and inference) time. Importantly, this speed-up is achieved by more efficiently using modern hardware and does not involve approximations like previous techniques (e.g. Linformer, Performer).

FlashAttention effectively nullifies the memory penalty for DimeNet++ models trained on QMugs (only an 1.68% increase) and also reduces the increase in training time to 2 minutes and 18 seconds.

Currently, FlashAttention is supported by a set of recent hardware architectures and is in prototype stage, so we still recommend the standard attention implementation for most practical use cases. However, we will update our repository as the technology matures.

It should also be noted that other than FlashAttention, another currently-available efficient implementation of attention scales with the square root of the number of atoms, at the cost of increased training time. Both FlashAttention and the memory-efficient variant are cited in our revised manuscript.

3. Scalability to larger systems

Although the standard attention implementation has quadratic scaling in the number of atoms, the models that we are targeting with our current work were not designed to work with molecules larger than the typical small, organic compound, or slightly larger variants (e.g. QMugs with pharmacologically relevant molecules).

We have found that the main computational bottleneck in our models is the underlying model itself. For example, we were unable to train DimeNet++ models on QMugs with the same batch size (128) as for the other datasets, even when using standard pooling functions like sum. Even a model with a batch size of 64 consumed 27.10 GB, which is close to the memory limit (32GB) of most commercially-available graphics cards, except the latest generation from NVIDIA which is prohibitive in terms of cost and not largely available. Well-known models such as DimeNet and sGDML exhibit quadratic scaling with respect to properties such as the number of atoms in the system or the number of neighbours per atom (more details in reference [3]). In our experience, SchNet models (implemented using schnetpack) are also problematic to run on GPUs even with small batch sizes (i.e. < 64) and inexpensive pooling operations like OWA on the OE62 dataset (62K molecules with up to 174 atoms).

However, ABP implementations like the ones described above should ensure that the proposed methodology is scalable to much larger systems. For the memory efficient variant, attention has been recorded to run for sequences of over 1 million in size (equivalent to number of atoms in for our tasks) with under 8 GB of memory.

Review 1

1. *In the introduction, the authors oversimplify the general strategies that have been taken for developing accurate, scalable and generalizable machine learning potentials. The ANI potential (ref. 8) and QM1 (ref. 9) models are mentioned as examples of the two classes that are highlighted. A couple of points that bear mention: fast, approximate QM methods such as NDDO-based semiempirical models or density-functional tight-binding methods can be formulated to have favorable (linear) scaling properties using linear-scaling electronic structure methods, or so-called quantum mechanical force field frameworks, e.g., J. Phys.: Condens. Matter (2017) 29, 383002-383016. This enables new QM/ML potentials to be developed that are more robust, do not have to be explicitly trained to avoid highly non-physical regions of configurational space, and that have certain known physics, such as long-ranged electrostatics, built into the models (it should be noted that the ANI potential of reference 8 does not, and breaks down when used for systems with varying charge). A drug discovery example is given in J. Chem. Phys. (2023) 158, 124110. While these points do not detract from the advances reported in this work, they do warrant placing into better context in the introduction.*

Answer: Thank you for the pointers to additional work. In our study, we restrict ourselves to improving only one class of methods that leverage variants of message-passing/graph neural networks which are typically purely data-driven, as opposed to the semiempirical methods suggested by the reviewer. We have clarified this in the introduction and added references to point readers toward this literature as well.

2. *The distinction between the current work and that of reference 21 should be better clarified beyond the statement "The current work represents an extension to previous work which considered only 2D molecular graphs, but demonstrated the potential of attention-based pooling for predicting properties such as the HOMO energy [21]."*

Answer: The relationship with our previous work has been detailed in the corresponding paragraph in the *Introduction*. In summary, previous work studied ABP in the context of various graph-learning benchmarks with traditional "2D" graph neural network methods (GCN, GIN etc). This included some molecular benchmarks but used connectivity/bonding and atom type information only. Notably, this did not attempt to encode spatial information (i.e. atomic coordinates), and as mentioned, accurate QML models typically require specialized models with geometric iso/equivariances, such as those studied here. Therefore, this work represents an exploration of learnable pooling functions in a new class of geometric models.

3. *It would be valuable to have tables in the SI that break down MAEs separately for training, validation and testing sets.*

Answer: The training and validation metrics have been added to the Supplementary Information (Supplementary Tables 1 and 2).

4. *To my understanding the architecture depicted in Fig. 1, which utilizes all atomic information in a system, is designed mainly for a small molecular systems, but may not be transferable to large systems. It would be valuable for the authors to please clarify this.*

Answer: We have clarified this aspect in the response letter and in the ‘Resource utilization and scaling’ subsection within the revised manuscript.

5. *In Section 5, the author claims that “Although the standard attention mechanism that we used here has quadratic time and memory scaling, we did not observe significantly larger training times or prohibitive increases in consumed memory for any of the shown experiments.” Is there any data to support this conclusion? Also, it is helpful to readers to give a detailed increased percentage.*

Answer: This is also addressed in the resource utilization subsection and at the beginning of this letter.

6. *In Eq.2, what does \top (T) represent? If \top is transpose, “ T ” is usually used to represent transpose, not \top .*

Answer: We have revised our notation for the transpose, which now uses the letter ‘ T ’. We initially opted for \top as the letter ‘ T ’ can potentially be confusing in the context of matrix names that involve capital letters encountered at the end of the alphabet (here Q , K , V).

7. *In Eq. 13, the meaning and usage of symbol α and β are not explained.*

Answer: The symbols denote global parameters that indicate the relative contributions of orbital energies and localisations to the loss. They are now explained immediately after Equation 13 in the manuscript.

8. *“Chen et al.” appears many times in the main text as well as the table captions, but the citation only appears when it appears the first time, making it hard to find this citation – it would be useful to have it listed at each occurrence.*

Answer: We have implemented this suggestion for the revised version.

Review 2

1. *The authors state that the proposed attention-based method does not significantly impact training times, which may hold true for the datasets considered in this work. However, it is important to note that atomic center-based machine learning potentials or machine learning force fields are generally designed to handle large size-extensive systems, allowing for simulations involving thousands to millions of atoms. Attention-based methods can potentially introduce additional computational costs for such large system, since quadratic scaling with the number of atoms N is problematic. It would be beneficial for the authors to address this aspect in the paper and provide a scaling test in terms of number of atoms present in system at least in inference stage, considering their potential impact on computational efficiency for larger-scale systems.*

Answer: We have addressed these concerns in this response letter as part of the ‘Resource utilization and scaling’ section at the beginning, as well as in the manuscript in the newly added subsection in *Results*.

2. *The authors have provided a description of the datasets and level of theory used in this study in the last paragraph of the introduction and results sections. While this information is crucial, it is equally important to discuss the choice of target properties and provide a more detailed elaboration on them. Since the aim of this work is to investigate local and general quantum mechanical properties, this will enhance the clarity and completeness of the manuscript.*

Answer: We have addressed this request in the revised version of the *Introduction*. We select a range of common quantum chemical properties that span intensive (e.g. excitation energies) and extensive (atomization energy) properties, including some that have specific atomic/spatial attribution (HOMO/LUMO) and those that are only definable on the molecular level (constitutional energies). These span applications in replication/acceleration of molecular simulations (total energies) to various catalytic and energy applications (frontier orbitals). In general, we have attempted to be diverse and comprehensive in terms of the properties studied while using only publicly available, standard datasets for easy comparison to other work.

3. *It would be valuable to further discuss the discrepancy improvement on DimeNet++ and ScheNet achieved by using Attention-Based Pooling (ABP) compared to sum pooling (in Figure 2). It is worth noting that in Chen et al. (DOI: 10.1039/d3sc00841j), a comparable improvement on GAP and ScheNet was shown in Figure 3 by using Ordered Weighted Averaging (OWA) pooling methods against sum pooling. Discussing and contrasting these findings can provide a comprehensive understanding of the interplay between proposed ABP and different machine learning models.*

Answer: We have added a discussion based on these ideas in the second paragraph of the *Discussion* section. In brief, using ABP tends to reduce the discrepancy between the models, which could be interpreted as SchNet, being a smaller and less expressive model, benefitting more from the increased flexibility offered by the ABP. This is particularly evident in the smaller datasets (QM7b, QM8), where it seems that the additional flexibility of ABP can offset the differences in the underlying model. Larger datasets such as QMugs, which also contain larger and more complex molecules, continue to show the benefit of the more expressive model (DimeNet++), even after adding ABP.

4. *The prediction of tensorial properties, e.g., dipole moment and molecular spectra, is another class of “general” properties in physics (arXiv:2102.03150, arXiv:2202.01449). Previous studies relied on equivariant representations and gated equivariant blocks to predict these properties. I suggest the authors should discuss the use of attention-based pooling in this direction and how it can be incorporated into equivariant neural networks as well.*

Answer: Attention-based pooling can be applied to any atomistic neural network by following the same recipe discussed in our work. In particular, this also applies to equivariant neural networks, as they still require an atom aggregation step as one of the last steps in the network. We have updated the *Introduction* to reflect this discussion.

5. *It would be informative to describe the technical differences and challenges between the current work and previous studies focusing on 2D molecular graphs. (<https://openreview.net/forum?id=yts7fLpWY9G>.)*

Answer: The relationship with our previous work has been detailed in the corresponding paragraph in the *Introduction*, as well as in this response letter as part of our answer to Reviewer 1, Comment 2. Previous work has only studied ABP in the context of traditional “2D” graph neural network methods (GCN, GIN etc), which did not attempt to encode spatial information (i.e. coordinates). As mentioned, accurate QML models typically require specialized models with geometric iso/equivariances, such as those studied here.

Review 3

1. *The title named 'local and general quantum mechanical properties' is unclear. In principle, some local properties still belong to general properties. Furthermore, this paper defined HOMO energy as a representative of localized properties. This is only half true. HOMO orbitals could be localized but also delocalized in these QM9, QM7 test datasets.*

Answer: We have added a clarification regarding the nature of the studied properties in the description of the datasets (*Introduction*), where we provide some more justification of the choices and language used. While HOMO orbitals can be delocalized, even over the entire extent of the molecule in question, they still retain a specific spatial/atomic localization (which may happen to be "all atoms"), and this localization pattern can and will vary between systems. It is this system-specific variation that we believe is not well described by averaging or summation over atomic representations. This can be contrasted with constituent energies which scale extensively.

2. *The authors explained the reason why they only focused on sum pooling. Nevertheless, I would highly suggest the authors to test the average pooling function as well based on extensive/intensive properties. Sum pooling is physically understandable for extensive properties in SchNet, which is based on local atomic interactions. However, the average pooling is commonly used for intensive properties even in the SchNet paper (J. Chem. Phys. 2018,148, 241722), where the HOMO energies in QM9 were calculated by using average pooling instead of sum pooling. Sum pooling usually leads to unphysical results for intensive properties. Although the authors claim that, based on their early work, the sum and average pooling have no big difference for quantum mechanic properties prediction, these tested properties are however almost extensive properties (such as U0,U298,CV,ZPVE,R2 refer to Comput. Chem. Eng. 2023,172.108202), which could make no big difference for sum/average pooling. This paper tested some intensive properties, the average pooling is therefore of high interest to compare for intensive properties, especially those in Figure 2 and Tables 1 and 4.*

Answer: We have included average pooling results for all the datasets and models which previously only had sum pooling results (QM7, QM8, QM9, QMugs, and MD17). These are included in the Supplementary Information (Supplementary Table 3) and discussed in the revised manuscript. Generally, we notice that average pooling can both improve or decrease performance compared to sum pooling, depending on the dataset and model. However, in our evaluation it still underperforms compared to ABP.

3. *Some long sentences are difficult to understand. For example, 'SchNet starts with atom embeddings based on the atom type and atom-wise layers implemented as linear transformations that are combined with convolutional layers satisfying rotational invariance'. This sentence means that the SchNet starts with atom embeddings based on atom type and atom-wise layers. However, the atom embedding is followed by the atom-wise layer instead of simultaneously processing. The other example sentence from the paper is 'The resulting Set Transformer module can be used as a pooling function by encoding ...'.*

Answer: We have clarified the highlighted instances for the revised version.

4. *For the Methodology part, please carefully check the equations. The parameter 'hu' seems to be 'ha' since u has never been introduced in this part. 'of dimension dk, dk, and dv' should be 'dq, dk, and dv'.*

Answer: We have reviewed the *Methodology* section and have fixed the definition of 'h_a' in Equation 1. The dimensions of the query, key, and value vectors are consistent with the intended meaning, namely that the queries and keys have the same dimension, as introduced in the original 'Attention is All you Need' paper. We have clarified this detail in the revised manuscript.

Furthermore, the Set Transformer's Multihead Attention Block (MAB) receives only two matrices as input (Q, K), such that the dimension d_k appears twice in the Linear modules of the MAB.

5. *The authors compared the ABP performance with OWA pooling. The RMSE of SchNet average pooling and OWA pooling in OE62 dataset are slightly higher than the results of Chen et al. (Chem. Sci.,2023,14,4913). One possible reason is perhaps that 1500 epochs used in this work overfitted the models. Please check the validation loss as the function of epoch to avoid overfitting.*

Answer: The differences that are observed compared to the published work of Chen et al. are expected, as we **(1)** generate 5 random different data splits to evaluate on and **(2)** we use a smaller batch size of 40 (Chen et al. used 50).

The 5 different random data splits allow us to perform a more robust evaluation, where we can report mean and standard deviation values. All the data splits maintain the same ratio of train, validation, and test samples as reported by Chen et al. This strategy is currently mentioned in the manuscript, and our splits are available in the GitHub repository.

The slightly smaller batch size is used to ensure that we do not encounter memory issues on our hardware (NVIDIA RTX 3090 with 24 GB VRAM and NVIDIA Tesla V100 with 16 and 32 GB VRAM), and that we do not have to rely on mixed precision features which might decrease learning performance.

Other factors that could influence the reported results are different software and hardware platform versions (e.g. using a newer version of Python, different operating systems, different GPUs and versions of CUDA toolkit, etc.). PyTorch does not guarantee completely reproducible results across different versions and platforms (<https://pytorch.org/docs/stable/notes/randomness.html>). Furthermore, the random initialisation of neural network weights and the random shuffling of training data in each epoch might be contributing factors.

Accounting for all factors, we consider that the results reported here are in agreement with the work of Chen et al., particularly the preprint which was available to us at the time of writing and conceptualising our study. Furthermore, Chen et al. report results with two decimal places, meaning that the exact values could be closer than they appear. Finally, the fact that for both OWA and average pooling we report slightly lower performance metrics supports the idea that our data splits are slightly harder

compared to the split of Chen et al, such that all evaluated methods will be slightly affected.

We do not expect overfitting to occur, as we use early stopping based on the validation loss for all methods reported in the paper. We do not set an explicit number of epochs in the code, and instead rely on early stopping to signal the right moment to finish training.

6. The OWA method incorporates the physical information in the model to make the model physically understandable. It can predict orbital energies as well as orbital locations simultaneously, which is useful for excitation and charge transfer studies of organic semiconductors. Other properties can also be predicted based on different physical references. Different properties should use different physical references for the atomic weights. Such as for LUMO energy prediction, LUMO orbital coefficients should be used for the model training. Using the HOMO orbital information model for the prediction of LUMO energies is doable but lacks physical sense.

Answer: We take the appropriate measures to evaluate OWA on LUMO correctly. More specifically, we use the information (coefficients, weights) available in the provided OE62 dataset, matching the training task. The code loads the matching data based on the task name (for example, HOMO or LUMO).

7. This learnable pooling function has a huge number of parameters, which could result in a demand for big data. It is interesting to investigate the learning curve of the APB performance with the increase of the training data compared with the sum or average pooling functions or the application for small dataset.

Answer: We have already benchmarked ABP against sum pooling (and mean pooling for the revised manuscript) for datasets with largely different sizes, in particular QM8 (21,786 molecules), QM7b (7,211 molecules) and MD17 (1,500 or 2,000 molecules depending on the dataset). These are all datasets with insufficient data points for deep learning standards, in particular for the five MD17 datasets. We observed that with the appropriate configuration (i.e. smaller number of heads and smaller embedding dimension per head), ABP generally outperforms both sum and mean pooling for these small datasets, or matches sum closely in the single occasion where sum pooling is better (Table 1 and Supplementary Table 3). Overall, the large number of parameters is intrinsic to standard attention; however, this does not imply overfitting or the inability to model small datasets, as shown by the evaluation.

8. The authors advertise their pooling method and mentioned that this pooling can deal with long-range interaction. Long-range interactions are challenging for current the state-of-the-art ML models. However, this should be tested such as in molecular dynamic simulations for some weak interaction systems like proteins. I would assume that the authors over advertise their method a bit as there is no test in the paper and it is not clear how this attention-based pooling could solve such problem.

Answer: We agree that our mention of long-range modelling was not sufficiently explained in the manuscript. We use the phrase ‘*potentially long-range or localised relationships due to the attention mechanism*’ in the paper to refer to an intrinsic property of the classical (as originally published) attention mechanism that is used. We have clarified this property here and in the manuscript.

In attention-based pooling we apply a Set Transformer on the atom (node) level outputs of a neural network. The Set Transformer consists of an encoder and a decoder, both leveraging the (usually classical) attention mechanism within modular blocks such as the Self Attention Block (SAB) and Pooling by Multihead Attention (PMA).

After the message passing steps (or equivalent), the learnt node representations are passed through the first SAB of the encoder. The classical attention mechanism considers all pairwise relationships between the learnt atom representations/ embeddings, regardless of the graph (molecule) connectivity information. This means that attention does not discriminate based on distance or position, and atom representation at extreme ends of the molecule can interact freely within the mechanism.

Furthermore, by stacking multiple SAB blocks within the encoder and the decoder (i.e., the outputs of one SAB become the inputs of the next SAB), the model is capable of learning higher-order relationships beyond just pairwise connections.

While we don't explicitly evaluate this property within our current work, we believe that the described property is worth highlighting in the context of standard pooling functions that do not have any mechanism to account for atom interactions at any distance.

We agree that it would make an interesting extension to study if this capacity to model long-ranged interactions can be translated into a demonstrable benefit in, for example, protein-protein interaction tasks. However, to our knowledge suitable benchmark datasets that can isolate these effects are not widely available.

9. The reference should be updated. There are some chemrxiv papers already published in journals.

Answer: We have now updated all references with their most up to date, published versions (where available).

10. This attention pooling method depends on the attention configuration (i.e. number of attention heads and hidden dimensions). This raises a question, how efficiently gets the optimal parameters and how expensive it is compared to the non-learnable pooling methods such as sum/average or other pooling functions.

Answer: We have covered several aspects related to this observation in this response letter and in the revised manuscript. As a general guideline, we have observed that the best ABP configurations tend to have large hidden dimensions for the SABs (512, 768, or 1024, noting that the embedding dimension per head is calculated as the hidden dimension divided by the number of heads, e.g. $1024 / 16$) and 12 or 16 attention heads. This applies especially for the larger datasets (QM9, QMugs, OE62).

Our measurements have shown that different ABP configurations have different impacts on the utilized resources (time and memory, Figure 3). In particular, configurations with lower attention values (hidden size, number of heads) consume similar amounts of resources to standard pooling, even with a standard implementation of attention (i.e. with quadratic scaling). Thus, experimentation with this kind of configurations comes at a minimal cost compared to standard pooling.

Lastly, we have also observed that models using ABP converge faster (in terms of number of epochs spent training). Overall, these observations support the idea that models using ABP can be used effectively for prototyping and hyperparameter search.

References

[1] Zaheer, M., Guruganesh, G., Dubey, K., Ainslie, J., Alberti, C., Ontanon, S., Pham, P., Ravula, A., Wang, Q., Yang, L., & others (2020). Big bird: Transformers for longer sequences. *Advances in Neural Information Processing Systems*, 33.

[2] David Buterez, Jon Paul Janet, Steven J Kiddle, Dino Oglic, & Pietro Lio (2022). Graph Neural Networks with Adaptive Readouts. In *Advances in Neural Information Processing Systems*.

[3] Musaelian, B. (2023). Learning local equivariant representations for large-scale atomistic dynamics. *Nature Communications*, 14(1), 579.

REVIEWERS' COMMENTS:

Reviewer #1 (Remarks to the Author):

The authors have done an adequate job at addressing the concerns of the previous review and has improved as a result. No further suggestions.

Reviewer #2 (Remarks to the Author):

The authors have sufficiently addressed the concerns and questions raised by the reviewers, resulting in an improved manuscript. I recommend that the manuscript is now suitable for publication.

Reviewer #3 (Remarks to the Author):

The quality of the paper has improved. I think the manuscript can be published now.